# Impact of Diffuse Idiopathic Skeletal Hyperostosis on Clinico-Radiological Profiles and Prognosis for Thoracic Ossification of Ligamentum Flavum-Myelopathy: A Propensity-Matched Monocentric Analysis

**DOI:** 10.3390/diagnostics12071652

**Published:** 2022-07-07

**Authors:** Baoliang Zhang, Guanghui Chen, Xi Chen, Zhongqiang Chen, Chuiguo Sun

**Affiliations:** 1Department of Orthopaedics, Peking University Third Hospital, 49 North Garden Road, Haidian District, Beijing 100191, China; zhangbl_spine@bjmu.edu.cn (B.Z.); chenguanghui88434@163.com (G.C.); chenxi93557@163.com (X.C.); 2Engineering Research Center of Bone and Joint Precision Medicine, 49 North Garden Road, Haidian District, Beijing 100191, China; 3Beijing Key Laboratory of Spinal Disease Research, 49 North Garden Road, Haidian District, Beijing 100191, China

**Keywords:** diffuse idiopathic skeletal hyperostosis, ossification of ligamentum flavum, thoracic myelopathy, prognosis, risk factor

## Abstract

Background: Diffuse idiopathic skeletal hyperostosis (DISH) has been evaluated as a potential risk factor of poor surgical outcomes for lumbar spinal stenosis, whereas the influence of DISH on neuroimaging characteristics and postoperative prognosis of patients with thoracic myelopathy has not been established. Therefore, this study aimed to shed light on this issue. Methods: A monocentric study enrolled 167 eligible patients with thoracic ossification of ligamentum flavum (TOLF), who were followed up for at least 2 years. Clinico-radiological parameters and surgical outcomes were compared between the DISH+ and DISH− groups before and after propensity matching. Subgroup analysis was conducted to compare the functional outcomes between mild DISH (M-DISH) and moderately severe DISH (MS-DISH) groups. Results: Fifty-eight patients were diagnosed as DISH, and its prevalence was 34.7%. Patients with DISH presented with older age, more males, taller stature, heavier weight, more commonly diffuse-type TOLF (*p* < 0.05). The DISH group showed significantly worse recovery rate (RR) at the final follow-up before and after propensity matching (*p* < 0.01), and slightly lower preoperative VAS, higher postoperative VAS and lower VAS reduction, despite not reaching the significant differences. Subgroup analysis demonstrated that the M-DISH group was associated with the lower mJOA score (*p* = 0.01) and RR at the final follow-up (*p* = 0.001), and tended to present higher preoperative VAS than the MS-DISH group. Conclusions: DISH has a significant predisposition to the elderly males with diffuse-type TOLF. Although the presence of M-DISH might bring about a suboptimal surgical outcome, both DISH and non-DISH patients experienced good neurological function improvements and pain relief through thoracic posterior decompression.

## 1. Introduction

Thoracic ossification of ligamentum flavum (TOLF) is the most prevalent contributor to thoracic myelopathy (TM), which mainly involves the Asian population [1]. The majority of TOLF patients present with an insidious and progressive natural history, and exhibit neurologic impairments in the middle-aged and elderly [2]. To block the symptomatic deterioration and restore neurological functions, surgical intervention is the only effective procedure for TOLF such as laminectomy with or without fusions, though unsatisfactory surgical outcomes and multiple perioperative complications remain visible [3]. Substantial evidence has demonstrated that multiple preoperative clinical and radiologic indicators could predict prognosis of TOLF patients treated surgically, such as age, disease duration, preoperative poor neurological status, and high signal intensity on T2-weighted MRI [4,5,6]. However, no consensus has been reached about predictors associated with the prognosis of TOLF.

Diffuse idiopathic skeletal hyperostosis (DISH) is a noninflammatory condition with the hallmark of progressive calcification and ossification initiating most frequently in the anterolateral aspect of lower thoracic spinal segments, and later extending into the upper thoracic segments and lumbar spine, which occasionally triggers localized back pain, spinal stiffness and ankylosing spinal fractures [7]. In addition, several studies have identified DISH as a potential predictor of discouraging prognosis for lumbar spinal stenosis (LSS) [8,9,10]. Yamada et al. reported that DISH extending to the lumbar segment was independently associated with unfavorable outcomes and reoperation for LSS [8]. Furthermore, Nakajima et al. elucidated that lumbar spinous process-splitting laminectomy at a lower segment adjacent to L-DISH made for further surgical treatment [10]. However, there has been little research investigating the impact of DISH on surgically treated patients with degenerative TM, especially TOLF.

A previous report indicated that DISH might be a potential cause in the induction of thoracic spondylotic myelopathy [11]. Clinically, there is a more common phenomenon that TOLF was frequently accompanied by DISH. Considering ossification lesions often occur at the lower thoracic segments and thoracolumbar junction, mechanical stress is thought to play an important role in the occurrence and development of TOLF. Presumably, mechanical stress alterations caused by consecutive vertebral bone bridges due to DISH are likely to be associated with TOLF pathogenesis. Moreover, we speculated that the stress at the responsible level or adjacent segment of DISH would affect their baseline neurologic status and postoperative functional recovery for TOLF patients. Hence, we will take the lead to lift the veil on the clinical relevance of DISH and TOLF through a retrospective monocentric study, including investigating the impact of DISH on clinico-radiological manifestations of TOLF patients, and exploring whether and how DISH negatively affects postoperative prognosis through propensity score matching and subgroup analysis.

## 2. Materials and Methods

### 2.1. Inclusion and Exclusion Criteria

The inclusion criteria were as follows: (1) symptomatic TOLF, evidence of thoracic spinal cord compression on magnetic resonance imaging (MRI) and/or computed tomography (CT); (2) undergoing laminectomy alone or laminectomy with fusion; (3) complete medical records and operation notes to determine the presence of DISH; (4) no previous thoracic spine surgery; (5) a minimal two-year follow-up after surgery; (6) willing to sign informed consent form. Patients were excluded if they were diagnosed with spinal trauma, active infection, neoplastic spinal disease, rheumatoid arthritis, Scheuermann’s disease, skeletal fluorosis or ankylosing spondylitis (AS), and underwent simultaneous circumferential decompression surgery for accompanying thoracic ossification of posterior longitudinal ligament (OPLL) and/or thoracic disc herniation (TDH).

### 2.2. Study Design and Eligible Subjects

The study was approved by the Research Ethical Committee of the Peking University Third Hospital on the grounds of the Helsinki Declaration. All patients signed informed consent and were followed up after surgery by telephone or outpatient review. We retrospectively reviewed 218 consecutive patients with clinically and radiographically confirmed symptomatic TOLF who underwent posterior thoracic decompression between January 2017 to January 2019. We further excluded 51 patients that did not meet the above criteria. Finally, a total of 167 TOLF patients with an average follow-up period of 36.74 ± 8.19 months were enrolled (Figure 1). There were 96 males and 71 females, with an average age of 55.97 ± 10.77 years (range 25–75 years).

### 2.3. Diagnostic Criteria of DISH

The definitive diagnosis of DISH was determined according to the criteria by Resnick as follows: (1) the presence of contiguous ligamentous ossification involving three or more intervertebral disk levels with anterior or lateral bridging; (2) preserved intervertebral disc space; (3) absence of apophyseal joint ankylosis and sacroiliac joint fusion [12]. Ossification of each disc space level from C7 to L1 was assessed and then graded according to the Mata scoring system [13]. Each vertebral level was scored as: (0) no ossification; (1) ossification without bridging; (2) ossification with incomplete bridging; (3) ossification with complete bridging of the disk space.

### 2.4. Grouping Criterion

These patients were divided into two groups based on the existence of DISH, the DISH+ group and DISH− group. DISH+ group included 58 patients and DISH− group included 109 patients. Following propensity score matching, there were 58 patients each in the matched DISH+ group (mDISH+ group) and matched DISH− group (mDISH− group). The severity of DISH was also investigated in the DISH+ group for a sub-study. According to the previous report, the extension of the ossification was described by the ossification index (OS-index), which was defined as the sum of the vertebral body and intervertebral disc levels involved by ossification lesions [14]. In the present study, we redefined OA-index as the sum of the vertebral body and intervertebral disc levels involved by ossification of anterior longitudinal ligament. Additionally, each vertebral and adjacent lower intervertebral space were considered as one segment. Namely, we stratified the DISH patients into two subgroups according to the thoracic OA-index: the mild DISH group (M-DISH group, 4 ≤ OA-index ≤ 8) and moderately severe DISH group (MS-DISH group, 9 ≤ OA-index ≤ 13). In subgroup analysis, M-DISH group contained 30 patients, and MS-DISH group contained 28 patients.

### 2.5. Surgery Indications and Intervention

Surgical indications, procedures, and decompressive range were determined at the discretion of the same experienced surgeon. Subjects were treated by two different techniques of posterior thoracic decompression: laminectomy alone and laminectomy with posterior instrumentation and fusion. The choice of the mode of operation was based on multiple factors, such as the compressive pathology, the degree of the degeneration, spinal instability, the sagittal alignment of the thoracic spine, and the patient’s physical conditions. Of all included patients, 34 patients underwent laminectomy alone, while 133 patients were treated by laminectomy with posterior instrumentation and fusion.

### 2.6. Data Collection and Processing

Demographic information (e.g., age, sex, BMI), medical history (e.g., duration of symptoms, history of hypertension and diabetes, smoking and drinking history), radiographical parameters (e.g., ossification classification and distribution, spinal occupying ratio, intramedullary signal intensity), surgical information (e.g., operation time, estimated blood loss, the number of decompressive segments, complications), and other available data for all subjects were retrospectively collected from the medical records. All surgery-related events that occurred within 30 days of the operation were defined as perioperative complications, mainly including dural tear and cerebrospinal fluid leakage, neurologic deterioration, surgical site infection and others.

### 2.7. Neuroimaging Evaluation and Outcome Measurements

TOLF distribution was divided into the focal type and diffuse type (continuous and skipping type). TOLF location was classified into three types (unilateral, bilateral, and bridged) on axial CT scan, and TOLF morphology fell into two types (round and beak) on sagittal MRI. The spinal canal occupying ratio (SCOR) was assessed by the axial maximum compression degree of the thoracic spinal cord, which was calculated as (axial ossified mass area/spinal canal area) × 100% [15]. Intramedullary high signal intensity (IHSI) at the narrowest level of the spinal cord were evaluated using the following grading: grade 0, none; grade 1, light; grade 2, intense. Clinical outcomes were assessed before surgery and at the final follow-up using the modified Japanese Orthopaedic Association (mJOA) scoring for thoracic myelopathy (Table 1) and the visual analog scale (VAS) for pain or numbness severity from the chest to the toes. The JOA score recovery rate (RR) was determined by [(postoperative score − preoperative score)/(11 − preoperative score)] × 100%, and was classified into excellent (75% to 100%), good (50% to 74%), fair (25% to 49%), and poor (0% to 24%). Moreover, the achieved JOA score was also evaluated as (postoperative mJOA score − preoperative mJOA score).

### 2.8. Statistical Analysis

Continuous variables were presented as means ± standard deviation and categorical variables as proportions. Each independent variable between the DISH+ group and DISH− group was compared by the independent *t*-test for continuous variables and the χ^2^ test and Fisher’s exact tests for categorical variables, as appropriate for the data distribution. Additionally, patients were propensity matched (1 DISH+: 1 DISH−) by potential differential predictors of surgical outcome. Furthermore, the subgroup analysis of patients was also conducted according to the OA-index of DISH. All statistical analyses were performed using SPSS statistics software, version 22.0. (SPSS, Chicago, IL, USA). *p*-value < 0.05 was considered statistically significant.

## 3. Results

### 3.1. Prevalence and Radiological Distribution Characteristics of DISH

Of 167 TOLF patients, fifty-eight patients were diagnosed as DISH with the prevalence of 34.7%. In these DISH subjects, a total of 754 segments (C_7_T_1_-T_12_L_1_) might be located, whereas a total of 441 (57.3%) segments were identified. The proportion of ossification lesions involving the upper thoracic, middle thoracic, and lower thoracic vertebrae was 29.48%, 36.05%, and 34.47%, respectively (Figure 2A). Furthermore, T_8,9_ (94.8%) and T_9,10_ (91.4%) were the segments where DISH occurs mostly (Figure 2B). The average number of ossification lesions was 7.6, and when stratified by age, the average numbers of ossified lesions in the age groups <40, 40–49, 50–59, 60–69, and ≥70 years was 4.8 ± 1.1, 5.6 ± 1.5, 7.4 ± 3.2, 8.9 ± 2.4, and 9.4 ± 2.5, respectively (Figure 2C). According to Meta scoring, the proportion of I, II and III grade were 19.7%, 29.5% and 50.8%, respectively (Figure 2D).

### 3.2. Demographic Characteristics between the DISH+ and DISH− Group

As showed in Table 2, the age of the DISH+ group was significantly older than that of the DISH− group (*p* = 0.025). Significant differences were observed in gender (*p* = 0.027), height (*p* = 0.001), weight (*p* = 0.036). However, comparable results were noted in body mass index (BMI) (*p* = 0.423), duration of symptoms (*p* = 0.422), smoking history (*p* = 0.479), drinking history (*p* = 0.758). Regarding comorbidities, no significant differences were noted among frequencies of hypertension (*p* = 0.553), diabetes mellitus (*p* = 0.328), hyperlipidemia (*p* = 0.633), cardiac diseases (*p* = 0.436), cerebrovascular diseases (*p* = 0.756) between two groups.

### 3.3. Neuroimaging Parameters between the DISH+ and DISH− Group

Regarding the segment distribution, the DISH− group showed the more distinct bimodal distribution, which was consistent with the overall distribution of TOLF, but the two peaks in DISH+ group became relatively flat (Figure 3A). The average affected segments of DISH+ group patients were more than those of DISH− group (*p* = 0.003). On sagittal MRI, patients with DISH were more susceptible to the diffuse-type TOLF, whereas those without DISH presented with more focal-type ossification lesions (*p* = 0.006, Figure 3B). However, comparable results were noted in sagittal ossification morphology (*p* = 0.116), axial ossification location (*p* = 0.244), occupying ratio (*p* = 0.122), intramedullary signal intensity (*p* = 0.125), concurrent OPLL (*p* = 0.128), and concurrent TDH (*p* = 0.323) between two groups (Table 3).

### 3.4. Perioperative Information between DISH+ and DISH− Group

Of the surgical techniques, laminectomy with fusion was performed most frequently in both the DISH+ (77.6%) and DISH− group (80.7%). The choice of surgical procedures showed no statistically significant difference between two groups (*p* = 0.631). Compared with the DISH− group, patients with DISH tended to be operated on a greater number of levels on average (*p* = 0.013); correspondingly, they had longer operative times (*p* = 0.037) and more estimated blood loss (*p* = 0.017). There were no significant differences in the follow-up periods (*p* = 0.880) and length of stay (*p* = 0.207). Regarding the perioperative complications, no significant differences were noted in cerebrospinal fluid leakage (*p* = 0.086), neurologic deterioration (*p* = 0.275), surgical site infection (*p* = 0.679) (Table 4).

### 3.5. Surgery Outcomes between the DISH+ and DISH− Group before and after Propensity Matching

Comparable results were observed in preoperative mJOA (*p* = 0.603) between two groups; however, postoperative JOA scores at the final follow-up were significantly lower in the DISH+ group than those in the DISH− group (*p* = 0.027). The JOA score RR was lower in the DISH+ group than the DISH− group (*p* = 0.001), and the achieved JOA scores in the DISH+ group tended to be lower than the control group (*p* = 0.099). Both the DISH+ and DISH− group showed significant improvements in postoperative JOA scores (*p* < 0.001 and *p* < 0.001, respectively). In addition, no significant differences existed in the preoperative VAS score, postoperative VAS score, and VAS reduction between two groups. Both the DISH+ and DISH− group showed significant reductions in VAS score postoperatively (*p* < 0.001 and *p* < 0.001, respectively) (Table 5).

After propensity matching (1 DISH+: 1 DISH−), baseline characteristics were comparable between the mDISH+ and mDISH− group. The JOA score at the final follow-up (*p* = 0.068) and the achieved JOA score (*p* = 0.243) were still lower in patients with DISH, though these did not reach significant differences. However, the JOA score RR was significantly lower in the mDISH+ group than in the mDISH− group (*p* = 0.003). In addition, no significant differences were still noted in the VAS before surgery (*p* = 0.958), VAS at the final follow-up (*p* = 0.462) and VAS reduction (*p* = 0.587) between two groups (Table 5).

### 3.6. Subgroup Analysis of Clinical Outcomes between the M-DISH and MS-DISH Group

Of 58 patients with DISH, 30 patients were identified as the M-DISH group and 28 belonged to the MS-DISH group. The two groups did not show any significant difference in preoperative JOA scores. The functional outcomes assessed by the JOA score at the final follow-up (*p* = 0.010) and RR (*p* = 0.001) in the MS-DISH group were better than in the M-DISH group though the achieved JOA scores were similar between two groups (*p* = 0.594). In addition, we found the preoperative VAS in the M-DISH group was higher than the MS-DISH group though it did not reach significant differences (*p* = 0.113), but the VAS scores at the final follow-up between two groups reduced to a similar level through decompression surgery. This caused the VAS reduction in the M-DISH group to tend to be more than that in the MS-DISH at the final follow-up (*p* = 0.059) (Table 6).

## 4. Discussion

In the present study, overall prevalence of DISH was 34.7% in the enrolled subjects, and this was a larger ratio compared to those in the general population, which was estimated as 3.85~17.50%, which might imply a clinicopathologic relevance of these two pathologies [16,17,18,19,20]. In addition, we found more than 70% of ossification lesions of DISH occurred at the middle and lower thoracic region, and the most affected segment was T_8,9_, followed by T_9,10_. These findings were in agreement with several previous studies [16,17,21,22]. For instance, Fujimori et al. [17] and Nishimura et al. [23] reported the T_8,9_ and T_9,10_ were the most involved levels of DISH in the general population and patients with spinal diseases. Interestingly, we noted the mean number of ossified lesions in thoracic DISH increased with aging, which suggested an age-related progressive natural process of bridging osteophyte formation in DISH. In particular, Yaniv et al. developed a semi-quantitative scoring system for evaluating osteophyte progression of DISH, and found a mean progression of one DISH grade per 1.6 years [24]. In addition, Lofrese et al. [25] reported a multicenter experience that relevant symptoms caused by cervical DISH typically develop in a chronic fashion because aging played a role in determining extension of hyperostosis and severity of symptoms, which indicated the “age of DISH” counts more than patients’ age with timeliness of decompression being crucial in determining clinical outcome. They considered that targeted bone resections could be reasonable in elderly patients, while more extended decompressions should be preferred in younger patients [25]. Similarly, whether the age of thoracic DISH influences the surgical outcome deserves further investigation.

Multiple epidemiological studies have demonstrated that DISH patients were universally older and more likely to be males than non-DISH patients in different populations, whereas the differences between height, weight and BMI remain a matter of considerable debate [18,19,20,21,22,23,26,27]. Okada et al. [26] reported that the mean BMI and weight instead of height were significantly higher in the DISH+ group than those in the DISH− group in 327 consecutive subjects undergoing the health checkups. Another survey for municipally registered Japanese residents showed comparable results in BMI of the DISH+ and DISH− group [18]. In addition, Kagotani et al. [20] evaluated the prevalence of DISH in 1647 individuals and found the height, weight and BMI in the DISH+ group were higher than the DISH− group. However, the phenomenon that the DISH+ group had taller stature and heavier weight but similar BMI in the present study would be due to the higher proportion of males in the DISH+ group. Regarding the neuroimaging features, previous established evidence indicated a distinctive bimodal distribution mode of OLF in the thoracic region [1,2]. Consistently, our findings confirmed such a distribution signature that in all the patients, the highest and second highest peak were found at T_9__,10_ and T_3__,4_, respectively, and in the DISH− patients, the highest and second highest peak were found at T_10__,11_ and T_3__,4_, respectively. Interestingly, in DISH+ patients, the two peaks were not that obvious, and overall distribution became relatively flat, and we speculated that the presence of DISH would disturb the distribution of TOLF. Additionally, we first identified a higher percentage of diffuse-type TOLF coupling with a greater number of compressed segments in the DISH+ group, which implied that the presence of DISH might accelerate the extension and progression of TOLF. By this token, these findings suggested the potential interaction of the natural courses of these two diseases, which deserves in-depth investigation in future.

Considering that DISH predominantly occurs at the thoracic region, it seems to be an issue whether the effect from DISH-derived mechanical stress on the thoracic spine will adversely affect the postoperative recovery of TOLF patients. Nakasuka et al. demonstrated that DISH was a potential risk factor in the induction of thoracic spondylotic myelopathy, but no differences in surgery outcomes were noted between patients with and without DISH [11]. Gao et al. also found no significant differences in terms of both neurological outcomes and surgical variants in the DISH+ and DISH− patients who underwent one-stage posterior circumferential decompression for concomitant thoracic OPLL and OLF [27]. However, these studies included only a small sample size, which might contribute to the unconvincing results. Contrary to their results, we detected that patients with DISH had a significantly worse neurological improvement in surgical outcome at 2-year follow-up as measured by mJOA RR, while the RR of both groups reached an excellent level (80.78% vs. 90.14%) after propensity matching. However, the achieved mJOA did not reach the statistically significant difference. Therefore, we concluded that DISH did not adversely affect functional improvement after posterior decompression surgery for TOLF, and it did not enable patients to achieve an optimal recovery, to some extent.

Spinal pain and stiffness are generally considered as a common symptoms of DISH patients [7]. However, previous studies have reported a conflicting result between DISH and back pain [28,29]. Schlapbach et al. [28] did not find significant differences in back pain between subjects with and without DISH. Another study, by Holton et al., demonstrated back pain in patients with DISH was less than the control group. The authors speculated that spinal hyperostosis might also be protective for back pain [29]. The present study demonstrated that both groups obtained an excellent pain relief through surgical intervention. As can be noticed, the patients with DISH presented with the slightly lower preoperative VAS scores, higher postoperative VAS scores and lower VAS score reduction. We contemplated that naturally fusing DISH could increase the stability of the spine and thereby limit pain in patients, to a certain extent; however, the global stability of the spine might be impaired and residual local pain existed as a result of decompression and laminectomy. As a whole, the presence of DISH did not negatively affect postoperative pain relief after posterior thoracic decompression.

To try to explain these phenomena, a subgroup analysis was conducted when dividing patients with DISH into MS-DISH and M-DISH group. For one thing, the findings showed that the JOA score at the final follow-up and JOA score RR were higher in the MS-DISH group than that in the M-DISH group. Interestingly, when comparing that with the overall results, whether original or normalized, we found the MS-DISH group and DISH- group showed similar neurological functional recovery, but the M-DISH group was associated with worse outcomes than the DISH− group. These results indicated short-segment and/or discontinuous DISH were the major contributor to affecting the neurological function recovery. This is similar to the findings of a previous study showing that patients with discontinuous OALL had a significantly worse percentage of recovery, and the authors interpreted that concentration of mechanical stress existed when the OALL was present both rostrally and caudally to the OLF, as well as the addition of the micromotion occurred on the vulnerable spinal cord and even the essentially immobile thoracic spine segments [30]. In addition, we found the baseline VAS score in the MS-DISH group tended to be lower than the M-DISH group, which was consistent with the above explanation that longer-segment continuous DISH could increase the stability of the spine, thus offering protection from pain stimuli. On all accounts, nearly all patients experienced effective pain reductions through surgical intervention.

Several limitations should be noted in this study. First, this was a retrospective study, which inevitably brought about some selection bias. Secondly, differences in sample size and heterogeneity between groups were a major limitation, though we applied propensity score matching to match patients’ baseline characteristics to make these factors as close as possible. Thirdly, decisions about surgical methods and the number of decompression levels were made at the discretion and preference of the surgeons. Fourthly, only thoracic mJOA scores and VAS were applied for the assessment of surgical outcomes, due to the limits of patients’ compliance and follow-up methods. Finally, we have not reported the ossification types or whether the patients had DISH adjacent to or distant from responsible lesions of TOLF. In fact, we found nearly all patients had same-segment and/or adjacent-segment DISH due to its diffuse phenotype, and breaking this population down would not have provided enough distant-segment DISH patients for statistical analysis.

## 5. Conclusions

This is the first study to comprehensively evaluate the impact of DISH on clinico-radiological presentations and postoperative outcomes of TOLF patients following posterior thoracic decompression. The CT-based prevalence of DISH in TOLF patients (34.7%) was higher than in general population, and its ossification lesions more affected T_8_, T_9_ and T_10_. Thoracic DISH has a significant predisposition to the elderly males with diffuse-type TOLF. Both the DISH+ and DISH− patients experienced good neurological function improvements and pain relief through thoracic posterior decompression, although the presence of M-DISH might bring about a suboptimal surgical outcome. Nevertheless, even if symptomatic TOLF patients have concomitant DISH, surgeons should not hesitate to perform operations, but the possibility that postoperative recovery is somewhat affected by DISH should be kept in mind and carefully explained to the patients before surgery.

## Figures and Tables

**Figure 1 diagnostics-12-01652-f001:**
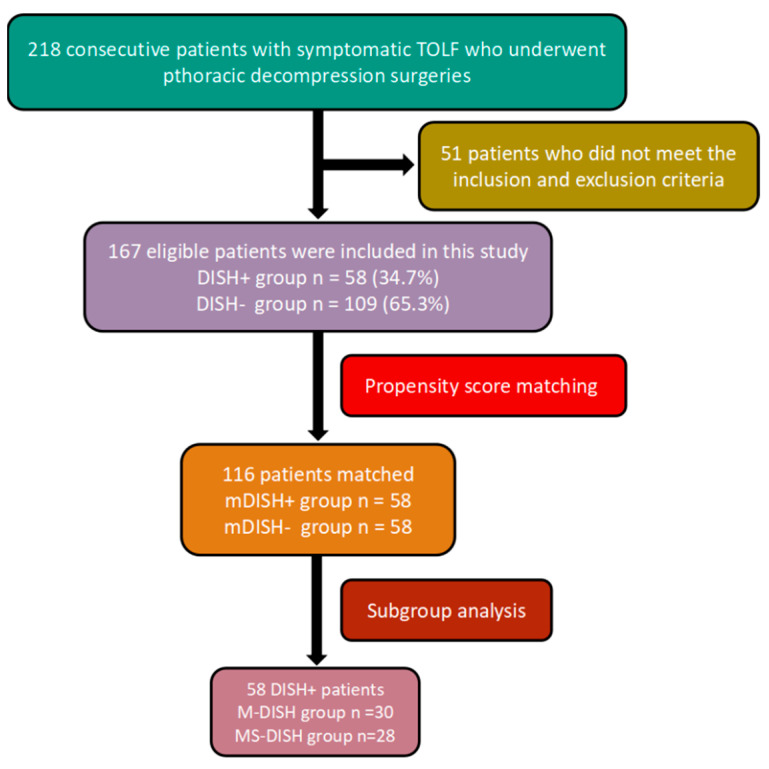
Flowchart of the study population. A total of 116 patients were selected by propensity score matching.

**Figure 2 diagnostics-12-01652-f002:**
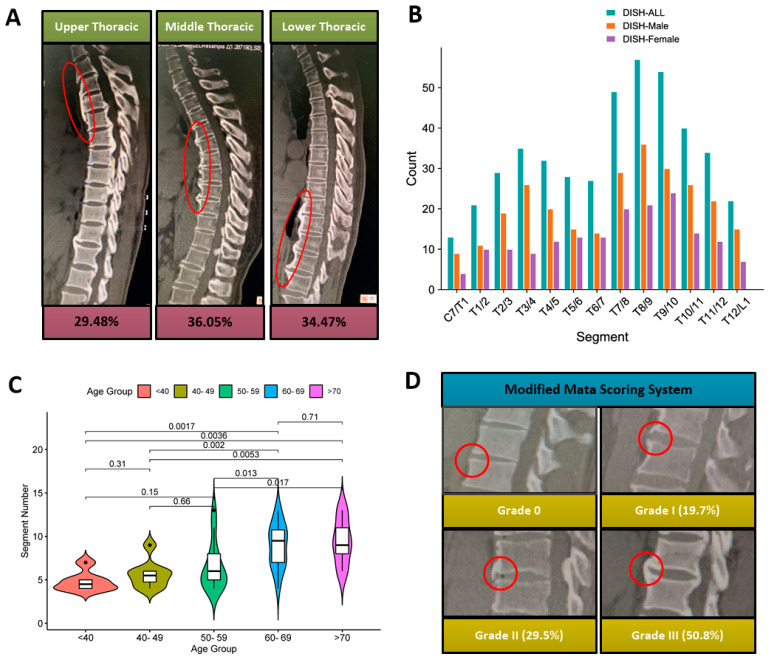
Distribution characteristics of DISH in TOLF patients. (**A**) Regional distribution. Ossification involving the upper, middle, and lower thoracic spine accounted for 29.48%, 36.05%, and 34.47%, respectively. (**B**) Segmental distribution. T_8,9_ (94.8%) and T_9,10_ (91.4%) were the segments where DISH occurs mostly. (**C**) The average numbers of ossified lesions might increase with age. (**D**) Severity distribution based on the Meta scoring, the proportion of I, II and III grade were 19.7%, 29.5% and 50.8%, respectively.

**Figure 3 diagnostics-12-01652-f003:**
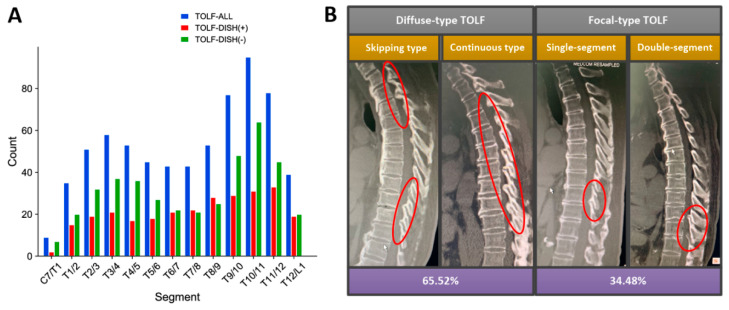
Distribution characteristics of TOLF between DISH+ and DISH− Group. (**A**) Segmental distribution. DISH− group showed distinct bimodal distribution, which was consistent with the overall distribution of TOLF, but the two peaks in DISH+ group became relatively flat. (**B**) DISH+ group was more susceptible to the diffuse-type TOLF (continuous-type and skipping-type) rather than focal-type TOLF (single-segment and double-segment).

**Table 1 diagnostics-12-01652-t001:** Modified Japanese Orthopedic Association (mJOA) scoring system for thoracic myelopathy.

Categories	Score (Points)
Impossible to walk	0
Need a cane or aid on flat ground	1
Need aid only on stairs	2
Possible to walk without any aid, but slow manner	3
Normal	4
Motor function: lower extremity	
Sensory function: lower extremity	
Apparent sensory disturbance	0
Minimal sensory disturbance	1
Normal	2
Sensory function: trunk	
Apparent sensory disturbance	0
Minimal sensory disturbance	1
Normal	2
Bladder function	
Urinary retention or incontinence	0
Severe dysuria (sense of retention, staining)	1
Slight dysuria (pollakisuria, retardation)	2
Normal	3
Total score	11

**Table 2 diagnostics-12-01652-t002:** Demographic characteristics of patients between the DISH+ and DISH− Group. DISH+ group contains 40 males while DISH− group contains 56 males. DISH+ group contains 40 males while DISH− group contains 56 males. Seventeen patients in DISH+ group has smoking history while 28 patients in DISH− group has smoking history. Seven patients in DISH+ group has drinking history while 15 patients in DISH− group has drinking history. Twenty-four patients in DISH+ group has hypertension while 40 patients in DISH− group has hypertension. Eleven patients in DISH+ group has diabetes mellitus while 28 patients in DISH− group has diabetes mellitus. Nine patients in DISH+ group has hyperlipidemia while 14 patients in DISH− group has hyperlipidemia. Thirteen patients in DISH+ group has cardiac diseases while 19 patients in DISH− group has cardiac diseases. Eight patients in DISH+ group has cerebrovascular diseases while 17 patients in DISH− group has cerebrovascular diseases.

Variable	DISH+ (*n* = 58)	DISH− (*n* = 109)	Statistical Value	*p*-Value ^1^
Age (y)	58.53 ± 10.72	54.61 ± 10.64	2.265	0.025
Males, no. (%)	40 (68.97)	56 (51.38)	4.792	0.029
Height (m)	1.69 ± 0.09	1.65 ± 0.08	3.371	0.001
Weight (kg)	77.74 ± 17.48	72.59 ± 13.48	2.114	0.036
BMI (kg/m^2^)	27.33 ± 7.16	26.64 ± 3.88	0.803	0.423
Disease duration (month)	28.36 ± 41.32	29.89 ± 42.53	−0.243	0.808
Smoking history, no. (%)	17 (29.31)	28 (25.69)	0.500	0.479
Drinking history, no. (%)	7 (12.07)	15 (13.76)	0.095	0.758
Hypertension, no. (%)	24 (41.38)	40 (36.70)	0.351	0.553
Diabetes mellitus, no. (%)	11 (18.97)	28 (25.69)	0.956	0.328
Hyperlipidemia, no. (%)	9 (15.52)	14 (12.84)	0.228	0.633
Cardiac diseases, no. (%)	13 (22.41)	19 (17.43)	0.607	0.436
Cerebrovascular diseases, no. (%)	8 (13.79)	17 (15.74)	0.097	0.756

Note: Quantitative data were presented as means with standard deviations (SD) and the counting data were presented as count with the percentage of the total in parenthesis. The ^1^ *p*-values comparing DISH+ and DISH− groups were determined using the Mann–Whitney test or chi-square test.

**Table 3 diagnostics-12-01652-t003:** Neuroimaging parameters of patients between the DISH+ and DISH− Group. TOLF distribution was divided into the focal type and diffuse type (continuous and skipping type). Total number of segments was defined as the number of ossification lesions of TOLF seen on CT or MRI. TOLF location was classified into three types (unilateral, bilateral, and bridged) on axial CT scan, and TOLF morphology fell into two types (round and beak) on sagittal MRI. Intramedullary high signal intensity (IHSI) at the narrowest level of the spinal cord were evaluated using the following grading: grade 0, none; grade 1, light; grade 2, intense.

Variable	DISH+ (*n* = 58)	DISH− (*n* = 109)	Statistical Value	*p*-Value ^1^
Ossification distribution (sagittal MRI)			7.599	0.006
Focal, no. (%)	20 (34.48)	62 (56.88)		
Diffuse (continuous/skipping), no. (%)	38 (65.52)	47 (43.12)		
Total number of segments, no. (%)	275 (40.50)	404 (59.50)		
Number of compressed segments	4.74 ± 2.72	3.71 ± 3.01	2.185	0.003
Ossification location (axial CT)			2.819	0.244
Unilateral, no. (%)	60 (8.84)	67 (9.87)		
Bilateral, no. (%)	133 (19.59)	202 (29.75)		
Bridged, no. (%)	82 (12.08)	135 (19.88)		
Ossification morphology (sagittal MRI)			2.467	0.116
Round, no. (%)	174 (25.63)	279 (41.09)		
Beak, no. (%)	101 (14.87)	125 (18.41)		
Occupying ratio	52.39 ± 22.93	55.03 ± 21.02	−1.549	0.122
Intramedullary signal intensity			4.164	0.125
Grade 0 (none), no. (%)	215 (31.66)	288 (42.42)		
Grade 1 (obscure), no. (%)	47 (6.92)	92 (13.55)		
Grade 2 (bright), no. (%)	13 (1.91)	24 (3.53)		
OPLL, no. (%)	30 (51.72)	43 (39.45)	2.318	0.128
TDH, no. (%)	10 (17.24)	23 (21.10)	0.979	0.323

Note: Quantitative data were presented as means with standard deviations (SD) and the counting data were presented as count with the percentage of the total in parenthesis. The ^1^ *p*-values comparing DISH+ and DISH− groups were determined using the Mann–Whitney test or chi-square test.

**Table 4 diagnostics-12-01652-t004:** Perioperative information of patients between the DISH+ and DISH− Group. Subjects were treated by two different techniques of posterior thoracic decompression: laminectomy alone and laminectomy with posterior instrumentation and fusion. Of all included patients, 34 patients underwent laminectomy alone, while 133 patients were treated by laminectomy with posterior instrumentation and fusion. All surgery-related events that occurred within 30 days of the operation were defined as perioperative complications, mainly including dural tear and cerebrospinal fluid leakage, neurologic deterioration, surgical site infection and others.

Variable	DISH+ (*n* = 58)	DISH− (*n* = 109)	Statistical Value	*p*-Value ^1^
Surgery procedures			0.231	0.631
Laminectomy with fusion, no. (%)	45 (77.59)	88 (80.73)		
Laminectomy alone, no. (%)	13 (22.41)	21 (19.27)		
Follow-up periods (month)	37.67 ± 8.91	36.24 ± 7.81	1.075	0.284
Length of stay (day)	8.88 ± 3.64	8.18 ± 2.65	1.268	0.207
Decompressed segments	5.37 ± 3.49	4.02 ± 3.27	2.504	0.013
Operation time (min)	132.76 ± 76.38	109.49 ± 63.46	2.099	0.037
Mean blood loss (mL)	552.41 ± 375.84	411.38 ± 452.84	2.404	0.017
Perioperative complications				
Dural tear and cerebrospinal fluid leakage, no. (%)	27 (46.55)	36 (33.03)	2.947	0.086
Neurologic deterioration, no. (%)	3 (5.17)	11 (10.09)	1.193	0.275
Surgical site infection, no. (%)	1 (1.72)	3 (2.75)	0.171	0.679
Others, no. (%)	2 (3.45)	3 (2.75)	0.063	0.802

Note: Quantitative data were presented as means with standard deviations (SD) and the counting data were presented as count with the percentage of the total in parenthesis. The ^1^ *p*-values comparing DISH+ and DISH− groups were determined using the Mann–Whitney test or chi-square test.

**Table 5 diagnostics-12-01652-t005:** Surgery outcomes of patients between the DISH+ and DISH− groups before and after propensity matching.

Variable	DISH+ (*n* = 58)	DISH− (*n* = 109)	Statistical Value	*p*-Value ^1^	mDISH+ (*n* = 58)	mDISH− (*n* = 58)	Statistical Value	*p*-Value ^1^
Preoperative mJOA	5.64 ± 2.33	5.44 ± 2.34	0.521	0.603	5.64 ± 2.33	5.53 ± 2.40	0.236	0.814
mJOA at the final follow-up	9.98 ± 0.98	10.31 ± 0.85	−2.229	0.027	9.98 ± 0.98	10.31 ± 0.93	−1.843	0.068
Achieved mJOA	4.34 ± 2.12	4.87 ± 1.84	−1.657	0.099	4.34 ± 2.12	4.78 ± 1.83	−1.173	0.243
mJOA recovery rate	80.78 ± 20.49	89.87 ± 11.10	−3.721	0.001	80.78 ± 20.49	90.14 ± 11.31	−3.044	0.003
Preoperative VAS	3.37 ± 3.74	3.45 ± 3.32	−0.141	0.888	3.37 ± 3.74	3.41 ± 3.35	−0.052	0.958
VAS at the final follow-up	0.91 ± 1.26	0.65 ± 1.04	1.440	0.125	0.81 ± 1.12	0.66 ± 1.15	0.738	0.462
VAS reduction	−2.47 ± 3.08	−2.81 ± 2.75	0.734	0.464	−2.47 ± 3.08	−2.76 ± 2.70	0.545	0.587

Note: Quantitative data were presented as means with standard deviations (SD). The ^1^ *p*-values comparing DISH+ and DISH− groups as well as mDISH+ and mDISH− groups were determined using the Mann–Whitney test.

**Table 6 diagnostics-12-01652-t006:** Functional outcomes between the M-DISH and MS-DISH group. MS-DISH group contains 18 males while M-DISH group contains 15 males.

Variable	MS-DISH (*n* = 28)	M-DISH (*n* = 30)	Statistical Value	*p*-Value ^1^
Age (y)	59.42 ± 9.26	57.70 ± 12.03	0.610	0.544
Males, no. (%)	18 (64.29)	15 (50.00)	1.205	0.272
Height (m)	1.69 ± 0.097	1.70 ± 0.082	−0.547	0.587
Weight (kg)	78.16 ± 14.96	77.35 ± 19.80	0.176	0.861
BMI (kg/m^2^)	27.36 ± 4.06	26.74 ± 6.26	0.446	0.658
Disease duration (month)	31.39 ± 40.88	25.53 ± 36.27	0.578	0.565
Preoperative mJOA	5.82 ± 2.55	5.47 ± 2.13	0.576	0.567
mJOA at the final follow-up	10.32 ± 0.76	9.67 ± 1.07	2.670	0.010
Achieved mJOA	4.50 ± 2.12	4.20 ± 2.14	0.536	0.594
mJOA Recovery rate	89.50 ± 12.51	72.65 ± 23.18	3.410	0.001
Preoperative VAS	2.57 ± 3.39	4.13 ± 3.95	−1.611	0.113
VAS at the final follow-up	0.89 ± 1.29	0.93 ± 1.26	−0.121	0.904
VAS reduction	1.68 ± 2.70	3.20 ± 3.27	1.924	0.059

Note: Quantitative data were presented as means with standard deviations (SD) and the counting data were presented as count with the percentage of the total in parenthesis. The ^1^ *p*-values comparing MS-DISH and M-DISH groups were determined using the Mann–Whitney test or chi-square test.

## Data Availability

Not applicable.

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
