# Peer review of "Impact of Diffuse Idiopathic Skeletal Hyperostosis on Clinico-Radiological Profiles and Prognosis for Thoracic Ossification of Ligamentum Flavum-Myelopathy: A Propensity-Matched Monocentric Analysis"

_diagnostics, 2022, doi:10.3390/diagnostics12071652_

Round 1

Reviewer 1 Report

The topic is very interesting. The study is well designed and the paper well written. I have a few minor concerns.

Discrete variables should be placed in median (range). This would be clearer

Demographic characteristics (section 3.1 and 3.2) should be placed in methods

Differences in sample size and heterogeneity between groups should be mentioned as a major limitation

Figure 2 B please place a higher quality image, highlighting TOLF

Author Response

Dear Editors and Reviewers:

Thank you for your letter and for the reviewers’ comments concerning our manuscript entitled “Impact of Diffuse Idiopathic Skeletal Hyperostosis on Clinico-Radiological Profiles and Prognosis for Thoracic Ossification of Ligamentum Flavum-Myelopathy: A Propensity-Matched Monocentric Analysis” (ID: diagnostics-1730347). Those comments are all valuable and very helpful for revising and improving our paper, as well as the important guiding significance to our researches. We have studied comments carefully and have made correction which we hope meet with approval. The main corrections in the paper and the responds to the reviewer’s comments are as flowing:

Responds to the reviewer’s comments:

  1. Response to comment: (……Discrete variables should be placed in median (range)……)

Response: We have added the range for each variable.

  1. Response to comment:(……Demographic characteristics (section 3.1 and 3.2) should be placed in methods……)

Response: As the title showed, we focused on the impact of DISH on the clinico-radiological profiles of TOLF, so demographic characteristics were important content of clinical parameters, which were compared between the DISH+ and DISH- group in the results section. In addition, we have placed general demographic characteristics in method part.

  1. Response to comment:(……Differences in sample size and heterogeneity between groups should be mentioned as a major limitation……)

Response: It is really true and valuable for these comments. Though we applied propensity score matching to match patients’ baseline characteristics to make these factors as close as possible, differences in sample size and heterogeneity between groups were still a major limitation, and we have mentioned it in the last paragraph of the discussion section.

  1. Response to comment:(……Figure 2 B please place a higher quality image……)

Response: We have changed a representative picture with a higher quality.

Reviewer 2 Report

After having carefully read the manuscript, I have some major concerns. Here my comments.

In methods section criteria for an additional fixation are a bit vague. How many patients in each group needed for a posterior arthrodesis and why? The authors should provide more specific indications on that. Moreover, I would suggest to restructure methods with a paragraph dedicated to the different groups analyzed, highlighting criteria for each different group (DISH and non-DISH group). On this field non-DISH group is not even mentioned in methods, while the analysis refers to it in results. The differentiation between patients undergone stand-alone laminectomy and those needing an adjunctive fixation should be somehow expressed even in terms of data reported in table 4. This aspect become a critical issue when considering outcome data. Why the authors put all together patients with and without fixations? The outcome should be predictably different regardless of DISH or non-DISH group considered. Similarly, the outcome could likely differ depending on the number of treated levels. M-DISH and MS-DISH with their respective descriptions and inclusion criteria are not mentioned in the methods, so it is difficult to understand what kind of patients have been sub-grouped and analyzed in results. The acronyms should be specified in the appropriate part of the paper and properly argued. Overall, the manuscript, although showing a wide mess of data, is not well structured, it is difficult to read, quite chaotic, and it is hard to perceive a clear novelty together with proper take home messages.

In the present form I wouldn’t consider this study for publication.  

Author Response

Dear Editors and Reviewers:

Thank you for your letter and for the reviewers’ comments concerning our manuscript entitled “Impact of Diffuse Idiopathic Skeletal Hyperostosis on Clinico-Radiological Profiles and Prognosis for Thoracic Ossification of Ligamentum Flavum-Myelopathy: A Propensity-Matched Monocentric Analysis” (ID: diagnostics-1730347). Those comments are all valuable and very helpful for revising and improving our paper, as well as the important guiding significance to our researches. We have studied comments carefully and have made correction which we hope meet with approval. The main corrections in the paper and the responds to the reviewer’s comments are as flowing:

Responds to the reviewer’s comments:

  1. Response to comment: (……In methods section criteria for an additional fixation are a bit vague. How many patients in each group needed for a posterior arthrodesis and why?……)

Response: As recommended, we have clearly elaborated the criteria for an additional fixation and the number of patients in each group. In addition, we are very sorry to find incorrect data presentation in the text when we rearranged the data. Of the surgical techniques, Laminectomy with fusion was performed most frequently in both the DISH (77.6%) and non-DISH group (80.7%) rather than laminectomy alone. As for the choice of surgical procedure, laminectomy with fusion was adopted for most patients to enhance their spinal stability. Only a few patients with single-segment or short-segment ossification, especially in upper thoracic spine, underwent the laminectomy alone. All these description has been added to the text.

  1. Response to comment:(……I would suggest to restructure methods with a paragraph dedicated to the different groups analyzed, highlighting criteria for each different group (DISH and non-DISH group)……)

Response: Considering the Reviewer’s suggestion, we have restructured methods with a paragraph dedicated to the different groups, highlighting criteria for each different group.

  1. Response to comment:(……This aspect become a critical issue when considering outcome data. Why the authors put all together patients with and without fixations? The outcome should be predictably different regardless of DISH or non-DISH group considered……)

Response: It is really true and valuable for these comments. Because this paper concentrated on the impact of DISH on the prognosis of TOLF patients, we also compared the adopted surgery procedures between DISH group and non-DISH group, and we found no significant difference between the two groups. Therefore, we considered the similar proportions of surgery procedures in each group would make the outcomes of DISH and non-DISH group comparable. In addition, some perioperative Information of DISH and non-DISH groups including the number of treated levels were significantly different, so we further used propensity score matching to match patients’ surgery-related factors to make these factors as close as possible. Therefore, the results after propensity matching will be more significant and referable.

  1. Response to comment:(……M-DISH and MS-DISH with their respective descriptions and inclusion criteria are not mentioned in the methods……)

Response: We are sorry that this part was not so clearly explained in detail. Considering the reviewer’ suggestions, we have further fully explained respective descriptions and inclusion criteria to make more understandable.

  1. Response to comment:(……The acronyms should be specified in the appropriate part of the paper and properly argued……)

Response: Considering the Reviewer’s suggestion, we have added the acronym at the back of the references in the text.

Round 2

Reviewer 2 Report

The paper seems improved reaching an adequate level. However, the discussion should be empowered referring to the most recent evidences in terms of DISH, especially in the paragraph “…Interestingly, we noted the mean number of ossified lesions in DISH increased with aging, which suggested an age-related progressive natural process of bridging osteophyte formation in DISH…”. This interesting aspect making the timeliness of diagnosis crucial for determining positive clinical outcomes has been nicely discussed in the paper on Global Spine J. 2021 Feb 16:2192568220988272. doi: 10.1177/2192568220988272. PMID: 33590802. I would suggest to stress this aspect, already mentioned for the cervical DISH, even for the thoracic DISH, considering your results in this sense. 

Author Response

Dear Editors and Reviewers:

Thank you for your letter and for the reviewers’ comments concerning our manuscript entitled “Impact of Diffuse Idiopathic Skeletal Hyperostosis on Clinico-Radiological Profiles and Prognosis for Thoracic Ossification of Ligamentum Flavum-Myelopathy: A Propensity-Matched Monocentric Analysis” (ID: diagnostics-1730347). Those comments are all valuable and very helpful for revising and improving our paper, as well as the important guiding significance to our researches. We have studied comments carefully and have made correction which we hope meet with approval. The main corrections in the paper and the responds to the reviewer’s comments are as flowing:

Responds to the reviewer’s comments:

Reviewer #2:

  1. Response to comment: (……However, the discussion should be empowered referring to the most recent evidences in terms of DISH, especially in the paragraph “…Interestingly……)

Response: It is really true and valuable for these comments. As recommended, the discussion section has been further enriched and empowered based on the most recent evidences in terms of DISH.

In addition, we have checked that all references are relevant to the contents of the manuscript again, and we have corrected and improved grammatical mistakes and redundant words/phrases and polished the expression of the article by using professional English editing.

We tried our best to improve the manuscript and made some changes in the manuscript. These changes will not influence the content and framework of the paper.

We appreciate for Editors/Reviewers’ warm work earnestly, and hope that the correction will meet with approval.

Once again, thank you very much for your comments and suggestions.

Zhongqiang Chen

Peking University Third Hospital